# Phase II trial of cisplatin, gemcitabine and pembrolizumab for platinum-resistant ovarian cancer

Christine S. Walsh [1,2¤a] *, Mitchell Kamrava[2,3], Andre Rogatko[2,4], Sungjin Kim[2,4], Andrew Li[1,2], Ilana Cass[1,2¤b], Beth Karlan[1,2¤c], Bobbie J. Rimel[1,2]

1 Department of Obstetrics and Gynecology, Division of Gynecologic Oncology, Cedars-Sinai Medical Center, Los Angeles, California, United States of America, 2 Cedars-Sinai Cancer, Cedars-Sinai Medical Center, Los Angeles, California, United States of America, 3 Department of Radiation Oncology, Cedars-Sinai Medical Center, Los Angeles, California, United States of America, 4 Biostatistics and Bioinformatics Research Center, Cedars-Sinai Medical Center, Los Angeles, California, United States of America

¤a Current address: Department of Obstetrics and Gynecology, University of Colorado, Aurora, Colorado, United States of America
¤b Current address: Department of Obstetrics and Gynecology, Dartmouth-Hitchcock Medical Center, Lebanon, New Hampshire, United States of America
¤c Current address: Department of Obstetrics and Gynecology, David Geffen School of Medicine at UCLA, Los Angeles, California, United States of America
* christine.walsh@cshs.org

**Data Availability Statement:** All relevant data are within the manuscript.

**Funding:** Funded by Merck Investigator Studies Program (MISP) grant #52261 to C.W. http://engagezone.msd.com/ds_documentation.php. The

## Abstract

### Objective

To evaluate the combination of pembrolizumab, cisplatin and gemcitabine in recurrent platinum-resistant ovarian cancer.

### Methods

Patients received six cycles of chemotherapy with gemcitabine and cisplatin on day 1 and day 8 of a 21-day treatment cycle. Pembrolizumab was administered on day 1 of cycles 3–6 and as maintenance monotherapy in cycles 7–34. Palliative radiation to a non-target symptomatic lesion was allowed. The primary objective was overall response rate by RECIST 1.1 criteria. Secondary objectives included safety, progression-free survival, time to progression, duration of response and overall survival.

### Results

An interim analysis for futility was performed at 18 evaluable patients. Overall response rate was 60%, duration of response was 4.9 months and time to progression was 5.2 months. Progression-free survival at 6 and 12 months was 43% and 5%. Median progression-free survival was 6.2 months and median overall survival was 11.3 months. In all patients, CA125 levels reflected response and progression. There were no pseudoprogression events. After receiving palliative radiation during pembrolizumab maintenance, a patient with recurrent ovarian clear cell carcinoma had an exceptional and durable response that is ongoing for greater than 2 years. After consultation with the sponsor, based on the modest

funders had no role in study design, data collection and analysis, decision to publish, or preparation of the manuscript.

**Competing interests:** This study was funded by the Merck Investigator Studies Program (MISP #52261) (CW). URL: http://engagezone.msd.com/oncology.php. The funders had no role in study design, data collection and analysis, decision to publish, or preparation of the manuscript. This does not alter our adherence to PLOS ONE policies on sharing data and materials.

duration of response observed at the interim analysis for futility, the decision was made to close the trial to further accrual.

## Conclusions

The addition of pembrolizumab to cisplatin and gemcitabine did not appear to provide benefit beyond chemotherapy alone in patients with recurrent platinum-resistant ovarian cancer.

## Introduction

Epithelial ovarian cancer is the most lethal of the gynecologic malignancies and is typically characterized by advanced stage at initial diagnosis, high rates of relapse, progressively shorter lengths of remission and development of resistance to therapy. There are limited treatment options for platinum-resistant ovarian cancer which recurs within 6 months of last platinum-based chemotherapy and there is a critical need to identify more effective treatment strategies for this fatal disease.

Since 2011, there have been many Food and Drug Administration approvals for antibodies targeting immune checkpoints including CTLA-4, PD-1 and PD-L1. Immune checkpoint inhibitors can reverse an immunosuppressive mechanism utilized by tumors to evade immune detection and durable responses have been observed in a subset of cancers, potentially due to the mobilization of an anti-tumor immune response [1].

We sought to test the efficacy of an approach combining an immune checkpoint inhibitor with a chemotherapy backbone in patients with platinum-resistant ovarian cancer. Laboratory studies suggest that gemcitabine has the ability to reverse platinum resistance [2] and three clinical trials support the use of the cisplatin and gemcitabine combination in platinum-resistant ovarian cancer [3–5]. Across the three trials, the overall response rates range from 16% to 70%, but the durability of response appeared limited, ranging from 5.4 to 11 months.

The purpose of this trial was to determine if the addition of the anti-PD-1 monoclonal antibody pembrolizumab to cisplatin and gemcitabine chemotherapy would improve upon responses and durability of remissions in platinum-resistant ovarian cancer. At the time we designed our trial, a phase II study utilizing the anti-CTLA-4 antibody ipilimumab in combination with platinum-based chemotherapy in first-line treatment of advanced non-small cell lung cancer demonstrated improvement in progression free survival (PFS) only when ipilimumab was administered after two rounds of chemotherapy in a phased regimen [6]. Based on this data, we planned our trial with a similar phased design.

## Patients and methods

### Study design and participants

The PemCiGem trial was conducted as a single-center, open-label, single-arm investigator-initiated phase II study that enrolled patients with recurrent platinum-resistant ovarian, fallopian tube or primary peritoneal cancer with measurable disease per Response Evaluation Criteria in Solid Tumors (RECIST v1.1) [7]. Patients must have had at least one prior platinum-based chemotherapeutic regimen and relapsed less than 6-months from most recent platinum-based treatment. Progression on a non-platinum regimen was allowed if the patient was considered platinum-resistant to the last platinum-containing regimen. Prior cisplatin and gemcitabine treatment was allowed. Patients were 18 years of age or older, had an Eastern Cooperative

Oncology Group (ECOG) performance status of 0 or 1 and had adequate renal, hepatic and hematologic function within 28 days of treatment initiation. Key exclusion criteria included immunodeficiency, active autoimmune disease requiring systemic treatment in past 2 years, active tuberculosis, active Hepatitis B or C, Human Immunodeficiency Virus, central nervous system metastases, carcinomatous meningitis, history of pneumonitis, active infection requiring systemic therapy and prior therapy with an anti-PD-1, anti-PD-L1 or anti-PD-L2 agent. The study was approved by the institutional review board at Cedars-Sinai Medical Center and was posted at www.clinicaltrials.gov (NCT02608684). All participants provided written informed consent.

## Treatment

Participants received gemcitabine 750 mg/m$^2$ IV over 30 minutes and cisplatin 30 mg/m$^2$ IV over 60 minutes on day 1 and day 8 of a 21-day cycle for six total cycles. Pembrolizumab 200 mg IV was administered over 30 minutes on day 1 after chemotherapy during cycles 3 to 6 and as a maintenance monotherapy every 21 days from cycle 7 until criteria for treatment discontinuation were met. Discontinuation criteria included withdrawal of consent, confirmed radiographic disease progression, unacceptable adverse events, intercurrent illness that prevented administration of treatment or completion of trial treatment with maintenance pembrolizumab (28 additional maintenance cycles for approximately 2 years total treatment).

Chemotherapy dose modifications were based on review of adverse events and laboratory studies drawn prior to each cycle. Chemotherapy was held on day 1 of each cycle for ANC < 1500 cells/mm$^3$ and/or platelet count < 100,000/μl. During cycles 1–2 (chemotherapy only), treatment was held for low counts for a maximum of 3 weeks. During cycles 3–6 (chemotherapy + pembrolizumab combination), chemotherapy was held for low counts and pembrolizumab monotherapy was administered on schedule. Hematologic parameters were reassessed for possible chemotherapy administration on the subsequent day 8. On day 8 of each treatment cycle, treatment was eliminated for ANC < 1000 cells/mm$^3$ and/or platelet count < 75,000/μl and the patient was assessed for treatment on the subsequent day 1. Chemotherapy dose reductions to two-levels were permitted before withdrawing the patient from further chemotherapy. Gemcitabine was dose-reduced to 600 mg/m$^2$ and 450 mg/m$^2$ for the following events during the prior cycle: febrile neutropenia, ANC < 500 cells/mm$^3$, platelet count < 50,000/μl or grade 3 liver toxicity. Cisplatin was dose reduced to 25 mg/m$^2$ and 20 mg/m$^2$ for grade 2 or greater peripheral neuropathy, grade 2 or greater renal toxicity or grade 3 or greater nausea or vomiting that did not respond to optimization of antiemetic therapy. Patients with chemotherapy treatment delay of more than 3 weeks were taken off chemotherapy and allowed to continue on pembrolizumab monotherapy.

## Concomitant therapies

Participants were not allowed to receive chemotherapy, biologic therapy, investigational agents or immunotherapy not specified in the protocol. Radiation therapy to a symptomatic solitary lesion was allowed per investigator discretion. Live vaccines were prohibited. Systemic glucocorticoids were prohibited except to treat suspected immunologic adverse events or as antiemetic premedication prior to chemotherapy in patients with nausea or vomiting refractory to non-steroid antiemetic therapy.

## Study assessments

Tumor imaging with CT scan or MRI was performed at baseline and after every 2 cycles of chemotherapy and after every 3 cycles during the maintenance phase. Imaging was allowed

earlier than defined by protocol as clinically indicated per discretion of the principal investigator. Serum CA125 levels were drawn with each cycle.

The primary endpoint was efficacy as defined by the overall response rate (ORR) by RECIST v1.1. Secondary endpoints for efficacy included PFS at 6 and 12 months (proportion of patients who were alive and progression-free at timepoint), time to progression (TTP, time from treatment start to disease progression by RECIST), duration of response (DOR, time from response to disease progression by RECIST) and overall survival (OS, time from treatment start to date of death).

Patients with evidence of disease progression after cycle 2 were considered refractory to chemotherapy and were allowed to drop chemotherapy and continue on pembrolizumab monotherapy starting with cycle 3.

Patients with progressive disease (PD) by RECIST 1.1 after cycle 4 were classified as iUPD (unconfirmed progressive disease) by iRECIST criteria [8] and allowed to continue treatment and undergo reassessment 4 weeks or later to confirm PD if the following criteria were met: patient was clinically stable with absence of signs or symptoms indicating disease progression, no decline in ECOG performance status, no evidence of rapid disease progression and absence of progression at critical anatomic sites such as the spinal cord. If repeat imaging demonstrated stable disease (iSD), partial response (iPR) or complete response (iCR), the patient was allowed to continue on study treatment and undergo regularly scheduled imaging assessments. If imaging demonstrated confirmed progressive disease (iCPD), patients were taken off study.

Safety and tolerability of the regimen were determined by assessing the frequency and intensity of adverse events as defined by the Common Terminology Criteria for Adverse Events (CTCAE v.4).

Tumor PD-L1 expression was assessed by a validated immunohistochemistry (IHC) assay using the Merck mouse monoclonal antibody clone 223C for testing of formalin-fixed, paraffin-embedded tissue [9]. Interpretation of PD-L1 reactivity was performed by a board-certified pathologist by evaluating the percentage of tumor demonstrating membrane staining and evaluating the presence of a distinctive PD-L1 staining pattern at the tumor/stroma interface. A Modified Percent Score (MPS) was calculated ranging from 0–100, representing the overall percentage of tumor and tumor infiltrating mononuclear inflammatory cells that had membrane staining at low (1+) intensities or greater. An H-score was calculated ranging from 0–300, reflecting the percentage of cells staining at each of the following intensities: negative (<1), low (1+), moderate (2+), high (3+). IHC assays and scoring were performed at QualTek Molecular Laboratories (Goleta, CA).

## Statistical analyses

This was a one-arm, phase II trial that followed the classical two-stage design proposed by Simon in 1989 [10]. The primary efficacy endpoint was ORR defined as the proportion of patients who achieve a CR or PR by RECIST 1.1 criteria. A sample size of 25 patients was calculated as required to test the null hypothesis of a response rate of less than 11.1% against the alternative hypothesis of a response rate of > 30.0% at 5.0% one-tailed level of significance and 80% power. Subjects meeting withdrawal criteria prior to receiving pembrolizumab with cycle 3 were replaced. An interim analysis for futility was scheduled at 18 evaluable patients. If 2 or fewer patients with favorable response were observed when 18 patients were evaluable, then the null hypothesis was accepted and the trial terminated. The probability of early stopping under the null was 0.68 and under the alternative was 0.06. Survival functions for PFS and OS were estimated by the Kaplan-Meier method [11]. Analyses were performed using R package version 3.5.3.

## Study milestones

The concept for this study was sent to the Merck Investigator Studies Program (MISP) in 9/2014. The protocol was developed in 3/2015 and approved in 8/2015. The FDA issued an IND exemption in 10/2015 and the trial was posted on clinicaltrials.gov in 12/2015. In 2/2016, the trial was approved by the Cedars-Sinai Medical Center Institutional Review Board. Patients were enrolled between 2/2016 and 11/2018. Among 21 participants enrolled to the trial, 16 (76.2%) were identified among patients undergoing treatment and Cedars-Sinai and 5 (23.8%) were self-referred or referred by a physician from another institution. All research took place at Cedars-Sinai Medical Center. After consultation with the sponsor in 3/2019, the decision was made to close the trial to further accrual based on a futility analysis at 18 evaluable patients. The last patient discontinued trial therapy in 6/2019. Follow-up is reported through 11/2019. Fig 1 demonstrates a Transparent Reporting of Evaluations with Non-randomized Designs (TREND) flowchart for the trial.

# Results

## Patient characteristics

Between February 2016 and November 2018, 21 patients with platinum-resistant recurrent ovarian cancer were enrolled in the trial. Patient characteristics are summarized in Table 1. The majority of patients had an initial diagnosis of stage III high-grade serous ovarian cancer. Genetic

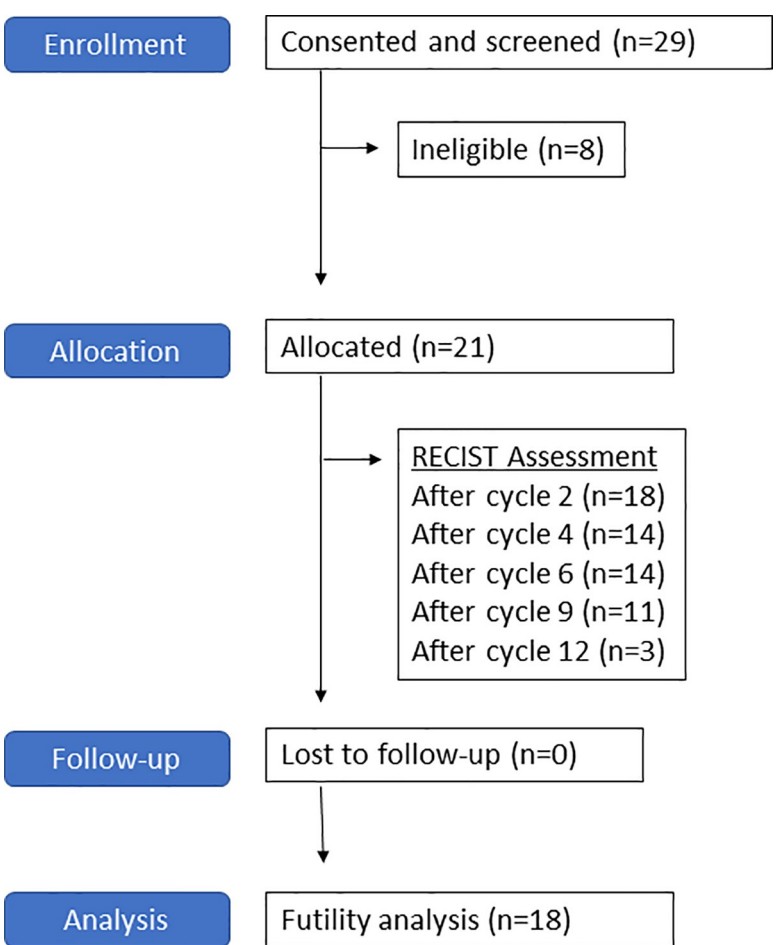

**Fig 1. TREND flowchart.** 21 eligible patients were treated per protocol. A futility analysis was performed when 18 patients were evaluable for response (RECIST assessment after cycle 2 of therapy).

**Table 1. Patient characteristics (n = 21).**

| Characteristic | n (%) |
|---|---|
| Median age, years (Range) | 55 (46–71) |
| FIGO Stage | |
| • I | 1 (4.8) |
| • II | 1 (4.8) |
| • III | 15 (71.4) |
| • IV | 4 (19.0) |
| Histology | |
| • Serous | 20 (95.2) |
| • Clear cell | 1 (4.8) |
| Site | |
| • Ovarian | 20 (95.2) |
| • Fallopian tube | 1 (4.8) |
| Genetic testing | |
| • BRCA1 | 1 (4.8) |
| • BRCA2 | 2 (9.5) |
| • BRIP1 | 1 (4.8) |
| • Negative/Unknown | 17 (80.9) |
| Treatment at primary diagnosis | |
| • Neoadjuvant chemotherapy/interval cytoreduction | 10 (47.6) |
| • Initial cytoreduction/adjuvant chemotherapy | 11 (52.3) |
| Recurrence history | |
| • Platinum sensitive, then platinum resistant | 9 (42.8) |
| • Platinum resistant at initial recurrence | 12 (57.1) |
| Number of previous lines of therapy | |
| • 1 | 2 (9.5) |
| • 2 | 11 (52.3) |
| • 3 | 6 (28.5) |
| • 4 | 1 (4.8) |
| • 5 | 1 (4.8) |
| Prior targeted therapy | |
| • Bevacizumab | 10 (47.6) |
| • PARP inhibitor | 4 (19.0) |

testing identified 4 carriers of deleterious germline mutations (19.0%): one *BRCA1*, two *BRCA2* and one *BRIP1*. At time of initial diagnosis, approximately half of patients had upfront cytoreduction followed by adjuvant chemotherapy (52.3%) and half underwent neoadjuvant chemotherapy with interval cytoreduction (47.6%). Nine patients initially had platinum-sensitive recurrences that progressed to platinum-resistant disease (42.9%) and twelve patients had platinum-resistant disease from initial recurrence (57.1%). The majority of patients enrolled had received two prior lines of chemotherapy (52.3%). Two patients enrolled after one prior line (9.5%) and two patients enrolled after four or five prior lines (9.5%). Ten patients had received bevacizumab (47.6%) and four patients had received a PARP inhibitor during a prior line of therapy (19.0%).

## Safety

Safety was evaluated separately for the different phases of treatment including chemotherapy (cycles 1–2: cisplatin and gemcitabine chemotherapy, n = 21), combination (cycles 3–6:

chemotherapy with pembrolizumab, n = 13) and maintenance (cycles 7–34, n = 12). Baseline symptoms and adverse events during the different phases of therapy are summarized in Table 2. Laboratory abnormalities are summarized in Table 3. Prior to receiving trial treatment, patients reported constipation (42.9%), pain (38.1%), dyspepsia (33.3%), fatigue (28.5%), nausea (28.5%), ascites (23.8%), hypertension (23.8%), bloating (19.0%), ileal obstruction (19.0%), insomnia (19.0%) and neuropathy (19.0%). These symptoms reflect those with ovarian cancer recurrence and prior therapies.

The most common adverse events of any grade during chemotherapy included nausea (61.9%), vomiting (47.6%), constipation (42.9%), fatigue (38.1%), pain (38.1%) and anorexia (23.8%). 4 patients (19.0%) were admitted to the hospital for small bowel obstruction (n = 2), congestive heart failure (n = 1) and intractable nausea and vomiting (n = 1). These hospital admissions reflected significant disease burden at the start of therapy. 3 patients (14.2%) developed thyroid function abnormalities despite not having yet started pembrolizumab therapy. Grade 3–4 laboratory abnormalities during chemotherapy included neutropenia (19.0%), hyponatremia (14.2%), lymphopenia (9.5%), anemia (4.8%) and hyperglycemia (4.8%).

Five patients discontinued trial therapy before reaching cycle 3 due to intercurrent illness related to ovarian cancer recurrence disease burden. An additional 3 patients had PD per RECIST 1.1 after cycle 2 and dropped subsequent chemotherapy and received single agent pembrolizumab only from cycle 3 onwards. A total of 13 patients went on to receive combination pembrolizumab and chemotherapy treatment. Among these patients, the most common adverse events of any grade included nausea (53.8%), pain (53.8%), fatigue (38.5%), headache (23.1%), hypertension (23.1%) and thyroid abnormalities (15.4%). Grade 3–4 laboratory abnormalities during the combination cycles included neutropenia (38.5%), thrombocytopenia (15.4%), anemia (7.7%), elevated AST (7.7%) and hyponatremia (7.7%).

A total of 12 patients continued onto the maintenance phase with pembrolizumab monotherapy. The most common adverse events of any grade during this phase included pain (50.0%), bloating (33.3%), fatigue (33.3%), thyroid abnormalities (25.0%), arthritis (16.7%) and neuropathy (16.7%). The only grade 3–4 laboratory abnormality was a single case of neutropenia (8.3%).

Only one patient discontinued therapy due to treatment-related adverse events. She discontinued therapy after 2 cycles of chemotherapy due to nausea and vomiting. No patients discontinued therapy due to toxicity attributable to pembrolizumab.

Fig 2 demonstrates the hematologic toxicities with impact on absolute neutrophil count (Fig 2A), platelet count (Fig 2B) and hemoglobin (Fig 2C) and the dose intensity delivered at each cycle of therapy from baseline to cycle 9 (Fig 2D). Rates of neutropenia were high, particularly on day 8 of cycles 3–6 (Fig 2A) and led to frequent dose reductions, dose delays and treatment holds per protocol (Fig 2D).

## Immune-related adverse events

A total of eight patients developed thyroid function abnormalities during trial treatment. 3 cases of hypothyroidism were observed prior to initiation of pembrolizumab at cycle 3. 5 patients developed thyroid abnormalities after initiation of pembrolizumab. We detected one case of central hypothyroidism/hypophysitis at cycle 3, one case of hyperthyroidism at cycle 4 that became hypothyroidism at cycle 5, 2 cases of hypothyroidism at cycle 8 and one case of worsening of preexisting hypothyroidism at cycle 10. One patient developed a grade 1 maculopapular rash on the bilateral lower extremities at cycle 27 that was treated with topical steroids and resolved by cycle 34. Among the 16 patients that received any pembrolizumab treatment, the immune-related adverse events included thyroid abnormalities (31.3%) and rash (6.2%).

**Table 2. Adverse events.**

| | Baseline | Chemo | | Combo | | Maintenance | |
|---|---|---|---|---|---|---|---|
| | n = 21 | n = 21 | | n = 13 | | n = 12 | |
| | Any Grade | Any Grade | Grade 3–4 | Any Grade | Grade 3–4 | Any Grade | Grade 3–4 |
| | n (%) | n (%) | n (%) | n (%) | n (%) | n (%) | n (%) |
| Admission | 0 | 4 (19.0) | 4 (19.0) | 0 | 0 | 0 | 0 |
| Anorexia | 3 (14.2) | 5 (23.8) | 2 (9.5) | 1 (7.7) | 0 | 0 | 0 |
| Anxiety | 2 (9.5) | 2 (9.5) | 0 | 2 (15.4) | 0 | 1 (8.3) | 0 |
| Arthritis | 0 | 1 (4.8) | 0 | 0 | 0 | 2 (16.7) | 0 |
| Ascites | 5 (23.8) | 2 (9.5) | 1 (4.8) | 0 | 0 | 1 (8.3) | 0 |
| Aspiration | 0 | 1 (4.8) | 0 | 0 | 0 | 0 | 0 |
| Bloating | 4 (19.0) | 3 (14.2) | 0 | 0 | 0 | 4 (33.3) | 0 |
| Chills | 0 | 0 | 0 | 2 (15.4) | 0 | 0 | 0 |
| Confusion | 0 | 1 (4.8) | 0 | 0 | 0 | 0 | 0 |
| Constipation | 9 (42.9) | 9 (42.9) | 0 | 1 (7.7) | 0 | 1 (8.3) | 0 |
| Cough | 1 (4.8) | 2 (9.5) | 1 (4.8) | 0 | 0 | 1 (8.3) | 0 |
| Dehydration | 0 | 1 (4.8) | 1 (4.8) | 0 | 0 | 0 | 0 |
| Diarrhea | 2 (9.5) | 2 (9.5) | 0 | 2 (15.4) | 0 | 1 (8.3) | 0 |
| Dizziness | 1 (4.8) | 1 (4.8) | 0 | 0 | 0 | 0 | 0 |
| Dry eye | 0 | 0 | 0 | 0 | 0 | 1 (8.3) | 0 |
| Dyspepsia | 7 (33.3) | 3 (14.2) | 0 | 0 | 0 | 0 | 0 |
| Dysphagia | 0 | 0 | 0 | 0 | 0 | 1 (8.3) | 0 |
| Dyspnea | 2 (9.5) | 1 (4.8) | 0 | 2 (15.4) | 0 | 1 (8.3) | 0 |
| Ear tingling | 0 | 0 | 0 | 0 | 0 | 1 (8.3) | 0 |
| Edema | 0 | 2 (9.5) | 1 (4.8) | 2 (15.4) | 0 | 1 (8.3) | 0 |
| Fatigue | 6 (28.5) | 8 (38.1) | 3 (14.2) | 5 (38.5) | 0 | 4 (33.3) | 0 |
| Gastroparesis | 1 (4.8) | 0 | 0 | 0 | 0 | 0 | 0 |
| Headache | 1 (4.8) | 3 (14.2) | 0 | 3 (23.1) | 0 | 0 | 0 |
| Heart failure | 0 | 1 (4.8) | 1 (4.8) | 0 | 0 | 0 | 0 |
| Hematoma | 0 | 1 (4.8) | 1 (4.8) | 0 | 0 | 1 (8.3) | 0 |
| Hemolysis | 0 | 0 | 0 | 1 (7.7) | 1 (7.7) | 0 | 0 |
| Hip fracture | 0 | 0 | 0 | 0 | 0 | 1 (8.3) | 0 |
| Hypertension | 5 (23.8) | 2 (9.5) | 1 (4.8) | 3 (23.1) | 1 (7.7) | 1 (8.3) | 0 |
| Hypotension | 0 | 1 (4.8) | 0 | 0 | 0 | 0 | 0 |
| Ileal obstruction | 4 (19.0) | 2 (9.5) | 1 (4.8) | 0 | 0 | 0 | 0 |
| Insomnia | 4 (19.0) | 3 (14.2) | 1 (4.8) | 1 (7.7) | 0 | 0 | 0 |
| Malaise | 0 | 1 (4.8) | 0 | 0 | 0 | 0 | 0 |
| Memory impairment | 0 | 0 | 0 | 1 (7.7) | 0 | 0 | 0 |
| Mucositis | 1 (4.8) | 0 | 0 | 1 (7.7) | 0 | 0 | 0 |
| Nausea | 6 (28.5) | 13 (61.9) | 0 | 7 (53.8) | 0 | 1 (8.3) | 0 |
| Neuropathy | 4 (19.0) | 1 (4.8) | 0 | 0 | 0 | 2 (16.7) | 0 |
| Neutropenic fever | 0 | 1 (4.8) | 1 (4.8) | 0 | 0 | 0 | 0 |
| Pain | 8 (38.1) | 8 (38.1) | 2 (9.5) | 7 (53.8) | 0 | 6 (50.0) | 0 |
| Palpitations | 0 | 0 | 0 | 1 (7.7) | 0 | 0 | 0 |
| Pleural effusion | 0 | 1 (4.8) | 0 | 0 | 0 | 0 | 0 |
| Pulmonary hypertension | 0 | 1 (4.8) | 0 | 0 | 0 | 0 | 0 |
| Rash | 0 | 0 | 0 | 2 (15.4) | 0 | 1 (8.3) | 0 |
| Tachycardia | 0 | 1 (4.8) | 0 | 2 (15.4) | 0 | 2 (16.7) | 0 |
| Thyroid | 2 (9.5) | 3 (14.2) | 0 | 2 (15.4) | 0 | 3 (25.0) | 0 |

*(Continued)*

**Table 2.** (Continued)

|  | Baseline | Chemo | | Combo | | Maintenance | |
|---|---|---|---|---|---|---|---|
|  | n = 21 | n = 21 | | n = 13 | | n = 12 | |
| Tinnitus | 0 | 1 (4.8) | 0 | 1 (7.7) | 0 | 0 | 0 |
| Upper respiratory | 0 | 1 (4.8) | 0 | 1 (7.7) | 0 | 0 | 0 |
| Urinary tract infection | 1 (4.8) | 0 | 0 | 1 (7.7) | 0 | 1 (8.3) | 0 |
| Vomiting | 3 (14.2) | 10 (47.6) | 2 (9.5) | 2 (15.4) | 0 | 0 | 0 |
| Weight loss | 1 (4.8) | 1 (4.8) | 0 | 0 | 0 | 0 | 0 |
| Weight gain | 0 | 2 (9.5) | 0 | 0 | 0 | 0 | 0 |

## Efficacy

Of the 21 patients that were enrolled and treated, 18 underwent imaging after cycle 2 and were evaluable for RECIST response. 14 had imaging after cycle 4, 14 after cycle 6, 11 after cycle 9 and 3 after cycle 12. Patients discontinued trial participation for intercurrent illness (n = 3), inability to tolerate chemotherapy (n = 1), progression (n = 16) and completion of 2 years of trial therapy (n = 1).

Among the 18 patients evaluable for RECIST response, best response was 1 CR (5.6%), 10 PR (55.6%), 5 SD (27.8%) and 2 PD (11.1%). The ORR was 61.1% and the clinical benefit rate (CBR) was 88.9%. The median duration of response among 16 patients achieving at least stable disease (months from first response to disease progression by RECIST 1.1) was 4.9 months (interquartile range 3.3 months). Fig 3A and 3B demonstrate best change from baseline in target lesion size. Fig 3C demonstrates response onset and duration.

**Table 3. Laboratory abnormalities.**

|  | Baseline | Chemo | | Combo | | Maintenance | |
|---|---|---|---|---|---|---|---|
|  | n = 21 | n = 21 | | n = 13 | | n = 12 | |
|  | Any Grade | Any Grade | Grade 3–4 | Any Grade | Grade 3–4 | Any Grade | Grade 3–4 |
|  | n (%) | n (%) | n (%) | n (%) | n (%) | n (%) | n (%) |
| Hemoglobin low | 4 (19.0) | 18 (85.7) | 1 (4.8) | 13 (100.0) | 1 (7.7) | 7 (58.3) | 0 |
| Platelets low | 2 (9.5) | 6 (28.5) | 0 | 7 (53.8) | 2 (15.4) | 3 (25.0) | 0 |
| ANC low | 0 | 10 (47.6) | 4 (19.0) | 10 (76.9) | 5 (38.5) | 4 (33.3) | 1 (8.3) |
| ALC low | 1 (4.8) | 10 (47.6) | 2 (9.5) | 4 (30.8) | 0 | 4 (33.3) | 0 |
| ALC high | 0 | 1 (4.8) | 0 | 1 (7.7) | 0 | 1 (8.3) | 0 |
| Albumin low | 0 | 2 (9.5) | 0 | 0 | 0 | 0 | 0 |
| ALT high | 1 (4.8) | 7 (33.3) | 0 | 7 (53.8) | 0 | 3 (25.0) | 0 |
| AST high | 2 (9.5) | 12 (57.1) | 0 | 10 (76.9) | 1 (7.7) | 7 (58.3) | 0 |
| Alk Phos high | 0 | 5 (23.8) | 0 | 1 (7.7) | 0 | 2 (16.7) | 0 |
| Calcium | 0 | 5 (23.8) | 0 | 1 (7.7) | 0 | 2 (16.7) | 0 |
| Creatinine high | 2 (9.5) | 2 (9.5) | 0 | 2 (15.4) | 0 | 3 (25.0) | 0 |
| Glucose high | 3 (14.2) | 5 (23.8) | 1 (4.8) | 1 (7.7) | 0 | 2 (16.7) | 0 |
| LDH high | 0 | 2 (9.5) | 0 | 1 (7.7) | 0 | 1 (8.3) | 0 |
| Magnesium low | 1 (4.8) | 4 (19.0) | 0 | 4 (30.8) | 0 | 3 (25.0) | 0 |
| Potassium low | 0 | 1 (4.8) | 0 | 0 | 0 | 0 | 0 |
| Sodium low | 0 | 3 (14.2) | 3 (14.2) | 1 (7.7) | 1 (7.7) | 1 (8.3) | 0 |
| Total protein high | 0 | 0 | 0 | 1 (7.7) | 0 | 0 | 0 |
| Uric acid high | 0 | 2 (9.5) | 0 | 1 (7.7) | 0 | 1 (8.3) | 0 |
| Urine blood | 1 (4.8) | 1 (4.8) | 0 | 1 (7.7) | 0 | 1 (8.3) | 0 |
| Urine protein | 0 | 1 (4.8) | 0 | 0 | 0 | 0 | 0 |

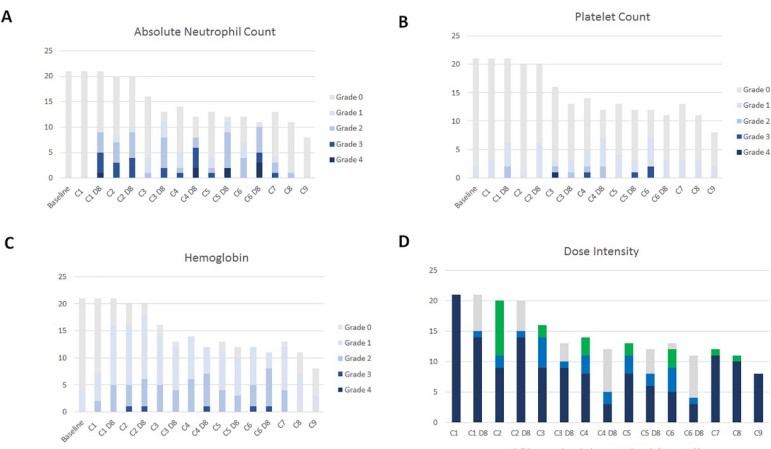

**Fig 2. Hematologic toxicity and treatment exposure per cycle among 21 patients who received trial treatment.**
Number of patients indicated on y-axis. Height of bars indicates the number of patients that are still receiving
treatment at each timepoint. Hematologic toxicities are graded per CTCAE version 4.0 and are indicated by colored
bars with darker colors indicating more severe toxicity in (A-C): (A) absolute neutrophil count; (B) platelet count; (C)
hemoglobin). (D) Summary of treatment exposure per cycle with number of patients receiving full dose (dark blue) or
undergoing dose reduction (light blue), dose delay (green) or treatment hold (grey).

Among all 21 patients on trial, the median TTP was 5.2 months (range 0.6–31.0 months).
PFS at 6 months was 42.9% and PFS at 12 months was 4.8%. Among 18 evaluable patients,
median PFS was 6.2 months (95% CI, 3.78–8.26) (Fig 3D). Among 21 patients on trial, 15 have
died. Median OS was 11.3 months (95% CI, 6.35–21.97 months) (Fig 3E).

## Pseudo-progression events

In this cohort, we observed no pseudo-progression events characterized by a tumor flare fol-
lowed by response. Five patients with iUPD that elected to continue therapy were confirmed
to have progressive disease on subsequent imaging studies and removed from trial.

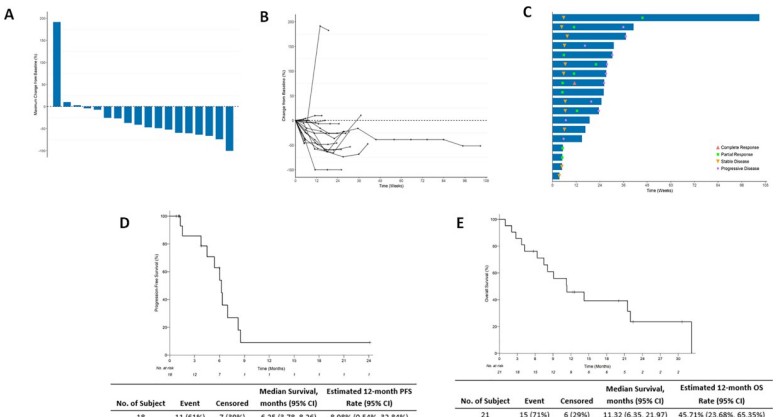

**Fig 3. Efficacy of cisplatin, gemcitabine, pembrolizumab combination.** (A) Waterfall plot demonstrating best
change from baseline in target lesion size. (B) Spider plot demonstrating best change in baseline in target lesion size
over time. (C) Swimmer plot demonstrating response onset and duration. Response changes at each timepoint are
indicated with symbols (CR with red triangle, PR with green square, SD with yellow inverted triangle, PD with purple
circle). (D) Kaplan-Meier curve of progression-free survival. (E) Kaplan-Meier curve of overall survival.

## CA125 response

We found CA125 response to accurately correlate with RECIST 1.1 response and symptom burden. We observed no cases where CA125 appeared to be falsely elevated due to an inflammatory response or tumor flare attributable pembrolizumab therapy. In all cases, CA125 levels appeared to drop with response and rise with progression. Among 18 patients with an initial CA125 response, all experienced subsequent rising levels. The reversal in CA125 trend occurred during the combination treatment phase in 5 patients and after completing cycle 6 of therapy in 9 patients. 4 patients discontinued study participation due to illness while CA125 levels were declining. Of concern, we observed a reversal in CA125 trend at the beginning of cycle 7 (just prior to the initiation of pembrolizumab monotherapy maintenance) in 6 patients and prior to cycle 8 in the remaining 3 patients.

## PD-L1 expression on tumor cells

Among 18 patients who received at least 1 cycle of pembrolizumab treatment, 14 had tumor evaluable for PD-L1 expression. One slide was assessed in 12 patients and two slides were assessed in 2 patients, totaling 16 slides assessed. Of the 16 evaluable samples, 12 (75.0%) demonstrated PD-L1 tumor staining greater than zero and presence of PD-L1 staining pattern at the tumor/stroma interface. The modified percent score (MPS) ranged from 0–50 (median, 1) and the H score ranged from 0–85 (median, 1). The MPS scores were overall low with the following distribution: MPS 0 (n = 4, 25%), MPS 1 (n = 5, 31.3%), MPS 2 (n = 3, 18.8%), MPS $\geq$ 5 (n = 4, 25%). One patient with more than one slide evaluated demonstrated discordant results at two omental tumor sites: omental site 1 (MPS 1 and H score 1); omental site 2 (MPS 25 and H score 50). There was no association between PD-L1 expression and efficacy of therapy.

## Impact of radiation therapy on response

Four patients received palliative radiation to a symptomatic non-target lesion at a dose of 8 Gray x 3 fractions. In all 4 cases, the radiation was administered as CA125 levels were rising and during pembrolizumab monotherapy. In 3 patients with recurrent high-grade serous carcinoma histology, all developed a rapid rise in CA125 levels following radiation treatment (Fig 4A). In

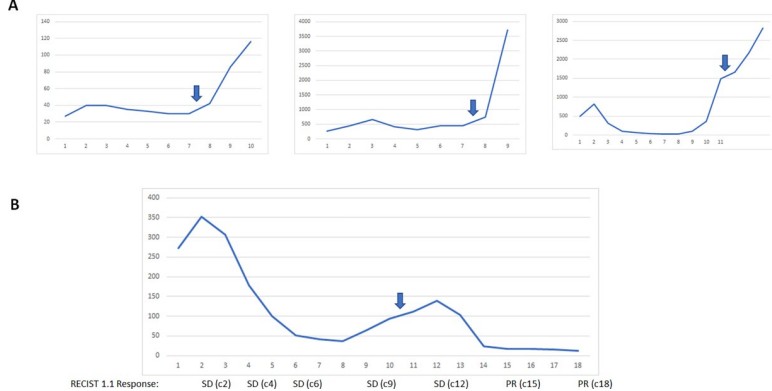

**Fig 4. Response in four patients who had rising CA125 levels during trial treatment and received palliative radiation to a non-target lesion.** Y-axis: CA125. X-axis: cycle number. Block arrow indicates timing of radiation administered at a dose of 8 Gray x 3 cycles. (A) CA125 trend in three patients with high-grade serous ovarian cancer demonstrates continuing rise in CA125 levels after receiving radiation treatment. (B) Following palliative radiation, one patient with clear cell ovarian cancer demonstrated decline and normalization of CA125 levels as well as shrinkage of lesions outside of the radiation field and conversion or RECIST 1.1 response from SD to PR, suggesting potential induction of an abscopal response.

one patient with recurrent stage IA ovarian clear cell carcinoma, radiation therapy led to a normalization of CA125 levels and shrinkage of tumor (Fig 4B). This patient had a symptomatic nodule in the left upper quadrant as CA125 levels were starting to rise after cycle 7. She received palliative radiation to this non-target lesion between cycles 10–11. CA125 started to decline after one additional cycle and normalized by cycle 14–15. A target aortocaval lymph node increased in size from 17 to 20 mm after radiation and subsequently shrank to 15 mm and then 11 mm on subsequent scans. She went on to complete 2 years of therapy and remains without evidence of disease 11 months after completing trial therapy. The conversion of her RECIST 1.1 response from SD to PR with shrinkage of lesions outside of the radiation field suggest that she may have experienced an abscopal response to the radiation and pembrolizumab combination. Immunohistochemistry studies demonstrated no PD-L1 expression and intact mismatch repair (MMR) protein expression in her tumor.

## Impact of addition of pembrolizumab to chemotherapy

An interim analysis for futility was performed at 18 evaluable patients in an intention-to-treat analysis. We confirmed efficacy with the primary trial end point with an ORR of 61.1%. However, the median duration of response of 4.9 months was modest. The addition of pembrolizumab to cisplatin and gemcitabine chemotherapy did not appear to provide benefit beyond the use of chemotherapy alone in this patient population (Table 4). After consultation with the sponsor, the decision was made to close the trial to further accrual.

## Discussion

In this single-center investigator-initiated trial evaluating a chemotherapy with immunotherapy combination in platinum-resistant ovarian cancer, we observed objective responses, but the addition of pembrolizumab to chemotherapy did not extend durability of response compared to prior studies evaluating the chemotherapy doublet alone. Nevertheless, this trial resulted in several important insights into the use of immune checkpoint inhibitors in ovarian cancer.

The combination regimen was tolerable. No patients discontinued treatment due to toxicity attributable to pembrolizumab. The rates of hematologic toxicity were highest during the combination phase and led to frequent dose reductions, treatment holds and delays. This impacted the overall dose intensity of the regimen. The rate of immune-related adverse events was low with 31% with thyroid abnormalities and 6% with rash. All immune-related toxicities were low-grade and manageable with standard therapies.

We did not observe any pseudo-progression events in our cohort. All patient with iUPD on imaging were confirmed to have iCPD on subsequent scans and discontinued treatment from the trial. Further, we found serum CA125 to be a reliable marker that correlated directly with

**Table 4. Clinical effect of gemcitabine + cisplatin combination with and without addition of pembrolizumab immunotherapy in platinum-resistant recurrent ovarian cancer.**

| Trial | N | ORR | CR | PR | SD | PD | PFS (m) | OS (m) | Response (m) |
|---|---|---|---|---|---|---|---|---|---|
| Rose 2003 | 35 | 15 (42%) | 4 (11%) | 11 (31%) | 9 (26%) | 11 (31%) | 6 | 12 | DOR: 11 |
| Nagourney 2003 | 27 | 19 (70%) | 7 (26%) | 12 (44%) | 7 (26%) | 1 (4%) | 6 | Not Reported | TTP: 7.9 |
| GOG-126L 2006 | 59 | 9 (16%) | 4 (7%) | 5 (9%) | 31 (54%) | 12 (21%) | Not Reported | 15 | TTP: 5.4 |
| PemCiGem 2019 | 18 | 11 (61%) | 1 (6%) | 10 (55%) | 5 (28%) | 2 (11%) | 6.3 | 11.3 | DOR: 4.9 TTP: 5.2 |

ORR = overall response rate; CR = complete response; PR = partial response; SD = stable disease; PD = progressive disease; PFS = progression-free survival; OS = overall survival; m = months; DOR = duration of response; TTP = time to progression.

radiographic response. We did not observe any false elevations in CA125 levels due to immunotherapy-related inflammation.

We had one exceptional responder after administration of palliative radiation during pembrolizumab monotherapy which led to reversal of a rising CA125 trend. A target paraaortic lymph node located outside the radiated field initially increased in size and then subsequently shrank, suggesting an abscopal systemic response. She completed 2 years of therapy and remains in remission 11 months after completing treatment. This durable response occurred in a patient with clear cell histology and provides support to a hypothesis from a prior trial suggesting clear cell ovarian cancer histology may be more responsive to immunotherapy approaches [12].

The use of palliative radiation did not lead to similar exceptional responses among 3 patients with recurrent serous ovarian cancer. In all 3 cases, the rate of CA125 rise appeared to accelerate after radiation. Further study is required to understand the basis for these disparate responses.

Our trial has several limitations. We designed our trial in March 2015 at a time when little was known about the responsiveness of ovarian cancer to immune checkpoint inhibitor therapy. We planned our protocol to allow a diverse patient population to enroll and have access to this therapy. We allowed for inclusion of patients with primary and secondary platinum resistance, treatment with an intervening non-platinum regimen, prior bevacizumab use and prior cisplatin and gemcitabine use. Many of our initially accrued patients had significant disease burden and intercurrent illnesses that led to hospitalizations, disease progression and discontinuation of chemotherapy. Administration of pembrolizumab monotherapy in these patients did not result in objective responses. Patients with higher baseline disease burden were less responsive and less tolerant to the protocol treatment. We have subsequently seen from a series of clinical trials that immune checkpoint inhibitor therapy has limited and modest activity epithelial ovarian cancer [13–17]. Further research is needed to identify the optimal ovarian cancer patients who may benefit from immunotherapy treatment.

In conclusion, the combination of gemcitabine, cisplatin and pembrolizumab treatment demonstrated tolerability and efficacy but lack of durability of response in patients with platinum-resistant recurrent ovarian cancer. The combination of palliative radiation and pembrolizumab may have led to an exceptional and durable abscopal response in a patient with recurrent MMR-proficient clear cell ovarian cancer.

## Supporting information

**S1 Checklist. TREND statement checklist.**
(DOC)

**S1 Protocol. Protocol version Feb 2016.**
(PDF)

**S2 Protocol. Protocol version Jan 2019.**
(PDF)

## Acknowledgments

We gratefully acknowledge all the patients that participated in this trial and their families. We thank the following site personnel for their clinical and regulatory support of the trial: Paula Anastasia, Meredith Axtell, Otilia Bilauca, Heather Chen, Sandra Contreras, Gillian Gresham, Emily Hautamaki, Leticia Jimenez, Sandra Lewis, Suwicha Limvorasak, Thanh-Toai Bethi Luu, Cynthia Martin, Melanie Ozaki, Katherine Rosenthal, Wendy Sabbah, Fay Shapiro,

Steven Stokes, Hang Tran. This trial is dedicated to Susan Petrovsky who inspired the development of this trial but lost her battle with ovarian cancer right before the trial was opened.

This study was presented at the Society of Gynecologic Oncology 50[th] annual meeting on Women's Cancer in Honolulu, Hawaii on March 19, 2019.

## Author Contributions

**Conceptualization:** Christine S. Walsh.

**Data curation:** Christine S. Walsh.

**Formal analysis:** Christine S. Walsh, Andre Rogatko, Sungjin Kim.

**Funding acquisition:** Christine S. Walsh.

**Investigation:** Christine S. Walsh, Mitchell Kamrava, Andrew Li, Ilana Cass, Beth Karlan, Bobbie J. Rimel.

**Methodology:** Christine S. Walsh, Andre Rogatko.

**Project administration:** Christine S. Walsh.

**Writing – original draft:** Christine S. Walsh.

**Writing – review & editing:** Mitchell Kamrava, Andre Rogatko, Sungjin Kim, Andrew Li, Ilana Cass, Beth Karlan, Bobbie J. Rimel.

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
