## [Decision Letter · Decision Letter 0]

1 Mar 2021

PONE-D-20-32083

Phase II trial of cisplatin, gemcitabine and pembrolizumab for platinum-resistant ovarian cancer

PLOS ONE

Dear Dr. Walsh,

Thank you for submitting your manuscript to PLOS ONE. After careful consideration, we feel that it has merit but does not fully meet PLOS ONE’s publication criteria as it currently stands. Therefore, we invite you to submit a revised version of the manuscript that addresses the points raised during the review process.

Please respond to the reviewers’ comments and submit a revised manuscript for consideration.

We look forward to receiving your revised manuscript.

Kind regards,

Sudeep Gupta, M.D.

Academic Editor

PLOS ONE

Additional Editor Comments:

Please revise the manuscript according to reviewers comments and resubmit.

Journal Requirements:

2. In your Methods section, please provide additional information about the participant recruitment method and the demographic details of your participants. Please ensure you have provided sufficient details to replicate the analyses such as: a) the recruitment date range (month and year), b) a statement as to whether your sample can be considered representative of a larger population, c) a description of how participants were recruited, and d) descriptions of where participants were recruited and where the research took place.

"Funded by Merck Investigator Studies Program (MISP) grant #52261 to C.W. http://engagezone.msd.com/ds_documentation.php.  The funders had no role in study design, data collection and analysis, decision to publish, or preparation of the manuscript."

We note that you received funding from a commercial source: Merck.

Reviewers' comments:

Reviewer's Responses to Questions

**Comments to the Author**

1. Is the manuscript technically sound, and do the data support the conclusions?

Reviewer #1: Partly

Reviewer #2: Yes

2. Has the statistical analysis been performed appropriately and rigorously? 

Reviewer #1: No

Reviewer #2: Yes

3. Have the authors made all data underlying the findings in their manuscript fully available?

Reviewer #1: Yes

Reviewer #2: Yes

4. Is the manuscript presented in an intelligible fashion and written in standard English?

Reviewer #1: Yes

Reviewer #2: Yes

5. Review Comments to the Author

Reviewer #1: The manuscript entitled ‘Phase II trial of cisplatin, gemcitabine and pembrolizumab for platinum-resistant ovarian cancer’ with the aim to evaluate the combination of pembrolizumab, cisplatin and gemcitabine in recurrent platinum-resistant ovarian cancer.

The manuscript requires further improvement.

Comments

Abstract

Page 3 Line 41, NCT02608684 to be omitted.

Statistical analysis

Page 10 Line 190-193, 1 or 2-tailed test to be stated. Attrition rates to be considered in the sample size calculation.

Page 11 Line 198-199, if most of the analysis were descriptive and no hypothesis testing done, the significance level not to be mentioned.

Results

All the tables require formatting according to the journal format.

Table 1 & 2, n to be replaced with N. At least one decimal point for the percentages Likewise with the figures in the text.

For Table 2, n (%) to be clearly denoted on top. 'Any G, G3-4, grade' to be clearly spelled out or denoted in the table footnote.

Table 2 & 3, the name for the column (first row) 4, 6 and 8 to be stated or combined.

Table 4 to be placed in results section. m to be denoted in table footnote.

Page 22 Line 326, median to be stated. Range to be replaced with IQR with a single value.

Figure 1, the flowchart requires improvement and to separate the allocation group and to be clearly structured according to the process.

Figures 2, 3 and 4 difficult to be visualized and requires enlargement.

Limitation of the study to be discussed.

References list presentation to follow journal format.

Reviewer #2: The authors provide us with a very well designed and carried out trial. It is evident that all the Ts have been crossed and Is dotted throughout the preparation and the conduction of this trial.

The issue of the addition of immunotherapy to this tough subgroup of patients with ovarian cancer is of utmost importance and I commend the authors on the effort.

The manuscript is very well written, the methods and results clearly explained and the discussion thorough.

A few questions and remarks:

1. was PDL1 tested on the tumors and if so can this be reported? If not - please discuss why not.

2. was MSI evaluated on the tumors in all patients or IHC for MMR proteins performed? If so can this be reported? If not - please discuss why not.

3. Discussion - 427-438: this paragagraph is a duplicate of a simillar paragraph in the results section. It is important for the discussion but should be shorter and more concise now that it is discussed.

6. PLOS authors have the option to publish the peer review history of their article (what does this mean?). If published, this will include your full peer review and any attached files.

Reviewer #1: No

Reviewer #2: No

---

## [Author Response · Author response to Decision Letter 0]

10 Apr 2021

Phase II trial of cisplatin, gemcitabine and pembrolizumab for platinum-resistant ovarian cancer

Response to Editor and Reviewer Comments

Journal Requirements:

The figures and supplementary files have been renamed to meet PLOS ONE’s style requirements. The figures have been renamed “Fig1.tif”, etc. The supplementary files have been renamed “S1 Protocol”, etc. 

2. In your Methods section, please provide additional information about the participant recruitment method and the demographic details of your participants. Please ensure you have provided sufficient details to replicate the analyses such as: a) the recruitment date range (month and year), b) a statement as to whether your sample can be considered representative of a larger population, c) a description of how participants were recruited, and d) descriptions of where participants were recruited and where the research took place.

We have included this additional information in our methods. The trial was posted on clinicaltrials.gov (track changes manuscript, page 11, line 221). The recruitment date range was from 2/2016 to 11/2018 (page 12, line 223). Participants were recruited from patients receiving care at Cedars-Sinai Medical Center or were self-referred or referred from their physician at another institution (page 12, line 224-226). All research took place at Cedars-Sinai Medical Center (page 12, line 226-227). 

"Funded by Merck Investigator Studies Program (MISP) grant #52261 to C.W. http://engagezone.msd.com/ds_documentation.php. The funders had no role in study design, data collection and analysis, decision to publish, or preparation of the manuscript."

We note that you received funding from a commercial source: Merck.

We have updated the Competing Interests Statement and have included it within the cover letter. 

Captions for Supporting Information files have been added at the end of the manuscript. There are no in-text citations to these materials. The supporting information files have been re-named to match the captions (track changes manuscript, p. 36). 

Reviewers' comments:

Reviewer #1: The manuscript entitled ‘Phase II trial of cisplatin, gemcitabine and pembrolizumab for platinum-resistant ovarian cancer’ with the aim to evaluate the combination of pembrolizumab, cisplatin and gemcitabine in recurrent platinum-resistant ovarian cancer.

The manuscript requires further improvement.

Comments

Abstract

Page 3 Line 41, NCT02608684 to be omitted.

Reference to the NCT number has been omitted (track changes manuscript, p. 3).

Statistical analysis

Page 10 Line 190-193, 1 or 2-tailed test to be stated. Attrition rates to be considered in the sample size calculation.

The statistical analysis description was updated to reflect use of a one-tailed test and the subject replacement strategy for any participants that discontinued trial participation prior to receiving pembrolizumab with cycle 3. Attrition rates were not otherwise considered in the sample size calculation (track changes manuscript, p. 11). 

Page 11 Line 198-199, if most of the analysis were descriptive and no hypothesis testing done, the significance level not to be mentioned.

Mention of the significance level has been omitted (track changes manuscript, p. 11). 

Results

All the tables require formatting according to the journal format.

All of the tables have been formatted according to the journal format. 

Table 1 & 2, n to be replaced with N. At least one decimal point for the percentages Likewise with the figures in the text.

In Tables 1 & 2, N has been replaced with n. Percentages have been updated to include one decimal point in the tables and in the text. 

For Table 2, n (%) to be clearly denoted on top. 'Any G, G3-4, grade' to be clearly spelled out or denoted in the table footnote.

“n (%)” has been added and “Any Grade” and Grade 3-4” have been spelled out at the top of tables 2 and 3. 

Table 2 & 3, the name for the column (first row) 4, 6 and 8 to be stated or combined.

In Tables 2 &3, the cells in the first row have been merged per the reviewer’s suggestion. 

Table 4 to be placed in results section. m to be denoted in table footnote.

Table 4 has been placed in the results section. “m” has been defined as “months” in the footnote (track changes manuscript, p. 28-29). 

Page 22 Line 326, median to be stated. Range to be replaced with IQR with a single value.

This line has been edited to clarify this is a median duration of response. The range has been replaced with the interquartile range (track changes manuscript, p. 24). 

Figure 1, the flowchart requires improvement and to separate the allocation group and to be clearly structured according to the process.

Figure 1 has been revised to clearly describe the allocation group and is structured according to the journal guidelines. 

Figures 2, 3 and 4 difficult to be visualized and requires enlargement.

Figures 2, 3, and 4 have been created as high-quality tif files that are at least 300 dpi resolution per journal formatting guidelines. 

Limitation of the study to be discussed.

We have modified our discussion to discuss the limitations of our study (track changes manuscript, p. 31-32). 

References list presentation to follow journal format.

The reference list presentation has been updated to Vancouver style to follow the journal format (track changes manuscript, p. 34-36). 

Reviewer #2: The authors provide us with a very well designed and carried out trial. It is evident that all the Ts have been crossed and Is dotted throughout the preparation and the conduction of this trial.

The issue of the addition of immunotherapy to this tough subgroup of patients with ovarian cancer is of utmost importance and I commend the authors on the effort.

The manuscript is very well written, the methods and results clearly explained and the discussion thorough.

A few questions and remarks:

1. was PDL1 tested on the tumors and if so can this be reported? If not - please discuss why not.

Tumor PD-L1 immunohistochemistry was performed, and this data are reported in the revised manuscript. 75% of evaluable tumors demonstrated PD-L1 expression. There was no association between PD-L1 expression and response to treatment. A paragraph was added to materials and methods (track changes manuscript, p. 10) and to results (track changes manuscript, p. 26). 

2. was MSI evaluated on the tumors in all patients or IHC for MMR proteins performed? If so can this be reported? If not - please discuss why not.

Mismatch repair IHC was not performed as part of the clinical trial protocol. We re-reviewed original pathology reports to collect data on clinical MMR IHC testing and found testing was done on one patient, our exceptional responder. The manuscript reports intact MMR IHC results for this patient (track changes manuscript, p. 27). 

3. Discussion - 427-438: this paragraph is a duplicate of a similar paragraph in the results section. It is important for the discussion but should be shorter and more concise now that it is discussed.

The paragraph in the discussion has been modified and shortened to avoid redundancy with the prior paragraph in the results section (track changes manuscript, p. 31). 

Additional revisions: 

Author affiliations have been updated (track changes manuscript, p. 1-2). 

Minor edits were made in the introduction (track changes manuscript, p. 5)

---

## [Editor Report · Decision Letter 1]

20 May 2021

Phase II trial of cisplatin, gemcitabine and pembrolizumab for platinum-resistant ovarian cancer

PONE-D-20-32083R1

Dear Dr. Walsh

We’re pleased to inform you that your manuscript has been judged scientifically suitable for publication and will be formally accepted for publication once it meets all outstanding technical requirements.

Kind regards,

Sudeep Gupta, M.D.

Academic Editor

PLOS ONE

Additional Editor Comments (optional):

The manuscript is now suitable for publication.
---

## [Editor Report · Acceptance letter]

24 May 2021

PONE-D-20-32083R1 

Phase II trial of cisplatin, gemcitabine and pembrolizumab forplatinum-resistant ovarian cancer 

Dear Dr. Walsh:

I'm pleased to inform you that your manuscript has been deemed suitable for publication in PLOS ONE. Congratulations! Your manuscript is now with our production department. 

Kind regards, 

on behalf of

Dr. Sudeep Gupta 

Academic Editor

PLOS ONE